# Predicting Emission Source Terms in a Reduced-Order Fire Spread Model—Part 1: Particulate Emissions

**Alexander J. Josephson** [1,*], **Troy M. Holland** [2], **Sara Brambilla** [3], **Michael J. Brown** [3] **and Rodman R. Linn** [1]

1 Earth and Environmental Sciences Division, Los Alamos National Lab, Los Alamos, NM 87545, USA; rrl@lanl.gov
2 Theoretical Division, Los Alamos National Lab, Los Alamos, NM 87545, USA; tholland@lanl.gov
3 Analytics, Intelligence, and Technology Division, Los Alamos National Lab, Los Alamos, NM 87545, USA; sbrambilla@lanl.gov (S.B.); mbrown@lanl.gov (M.J.B.)
* Correspondence: alexanderj@lanl.gov; Tel.: +1-505-665-6589

**Abstract:** A simple, easy-to-evaluate, surrogate model was developed for predicting the particle emission source term in wildfire simulations. In creating this model, we conceptualized wildfire as a series of flamelets, and using this concept of flamelets, we developed a one-dimensional model to represent the structure of these flamelets which then could be used to simulate the evolution of a single flamelet. A previously developed soot model was executed within this flamelet simulation which could produce a particle size distribution. Executing this flamelet simulation 1200 times with varying conditions created a data set of emitted particle size distributions to which simple rational equations could be tuned to predict a particle emission factor, mean particle size, and standard deviation of particle sizes. These surrogate models (the rational equation) were implemented into a reduced-order fire spread model, QUIC-Fire. Using QUIC-Fire, an ensemble of simulations were executed for grassland fires, southeast U.S. conifer forests, and western mountain conifer forests. Resulting emission factors from this ensemble were compared against field data for these fire classes with promising results. Also shown is a predicted averaged resulting particle size distribution with the bulk of particles produced to be on the order of 1 μm in size.

**Keywords:** fire simulations; particle emissions; surrogate modeling

## 1. Introduction

While interest in the cause and effects of wildfires has increased dramatically over the last few decades, researchers have struggled to predict fire behavior. Although several wildfire CFD models have been developed in recent years (LANL's FIRETEC [1], NCAR/UC-Boulder's WRF-Fire [2], NIST's FDS [3], etc.), with varying degrees of success, all have had a similar problem with the computational cost associated with simulating a full-sized wildfire. Recently, a reduced-order fire spread model, QUIC-Fire, was developed which significantly reduced the computational cost of wildfire simulations while maintaining an acceptable degree of accuracy in its predictive capabilities [4]. In its initial inception, QUIC-Fire focused on fire spread characteristics and did not predict emissions from fires. This work focuses on the development and implementation of a predictive model for the formation of particle emissions which could be implemented in QUIC-Fire.

Predicting the formation of particles in a wildfire situation has proven to be a difficult proposition for simulation softwares. While there exist physics-based models for predicting the behavior of particles post-combustion [5], to the authors' knowledge, there are no physics-based models for

predicting the initial formation and emission of particles yet implemented in any wildfire CFD model. Those CFD models which do contain a particle emission models are usually based on experimentally measured emission factors, a representative value which quantifies mass of particles emitted over mass of fuel consumed, read from a table, not predicted from the fundamental physics of the process. Emission factors for particle emissions are usually dependent loosely on fire intensity, fuel density, or fuel species [6] and generally have large associated uncertainties [7,8].

Implementation of a physics-based particle formation model in a wildfire CFD model is difficult for two reasons. First, there are very few models in existence which can predict this particle formation. Second, the fundamental physics and chemistry which govern the initial formation of soot, the backbone of wildfire particulate emissions, occur on a microscale level, and implementation of a fundamental model would require resolution of the simulation at a millimeter to centimeter scale. Most wildfire CFD models are resolved on the meter or kilometer scale as a resolution on the microscale is infeasible. In this study, we implemented a highly-detailed, physics-based model on a microscale in a variety of scenarios one might see in a wildfire. The results of these simulations were used to identify key characteristics of the fire which govern particle formation and then calibrate a proposed, easy-to-evaluate, surrogate model to these simulations. Experimental data were then used to validate the proposed surrogate model.

## 2. Model Development

Increasingly popular among wildfire models are a class of cellular automata models [9,10]. Among these cellular automata models are those which model fuel as a series of potential energy packets that may be activated through the combustion process [11]. When combustion occurs, these energy packets may react in a number of ways as governed by fundamental physics and thermodynamics. One way the packets may react is to jump locations and distribute heat, igniting another energy-packet and/or radiating heat to the atmosphere. In this work, we consider these jumps of energy as flamelets of a combustion process and envision fire simply as a cumulative agglomeration of these flamelets. In developing this particle emission source model, we wanted to answer the fundamental question: How much soot is produced by a single flamelet? In developing this model, we assumed the primary source of particle emissions is associated with the formation of soot particles and subsequently ignored ash and dust as negligible sources of mass in $PM_{2.5}$ [12], the particulate emissions smaller than 2.5 μm.

### 2.1. Flamelet Simulation

To determine the amount of soot produced by a flamelet, we first need to characterize the flamelet. To obtain the soot profile, we treated each flamelet as a one-dimensional line, as shown in Figure 1. At the starting location, there is just solid fuel which is heated-up and pyrolyzed. From this pyrolysis, gases, of a density determined by temperature and pressure, enter a fuel-rich zone. Consistent with a diffusive flame [13], gases in this region heat up and mix with counter-diffusing air. When the mixture reaches its stoichiometric point, where the chemical ratio is balanced between volatile fuel and oxygen according to a reaction of complete combustion, a reaction zone commences where most volatiles are oxidized. Lastly, there is a cool-off zone of 'atmosphere' where products of combustion are further diluted by air and the temperature decreases towards ambient. In these simulations, the flamelet length, or length of the energy packet's 'jump', was the length of the fuel-rich zone, the reaction zone is 1/4 the length of the flamelet length, and the atmospheric zone is taken as 1 m beyond the flamelet length, as this gave adequate time for particle agglomeration to asymptote.

To this characterized flamelet, we applied a detailed physics-based soot model [14] designed to predict soot formation from solid complex fuels, such as biomass. With regard to soot formation, there are three important pieces of information needed to carry out this flamelet simulation: a temperature profile, a gas chemistry profile, and an initial yield of soot precursors.

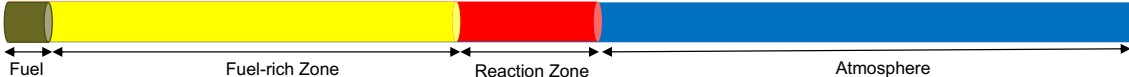

| Fuel | Fuel-rich Zone | Reaction Zone | Atmosphere |

**Figure 1.** Depiction of the flamelet simplification, a 1-dimensional evolution of a flamelet starting from fuel going through a primary pyrolysis reaction, fuel-rich heat-up zone, hot reaction zone, and last, a particle agglomeration zone in the atmosphere.

A temperature profile was obtained from the temperature profile of an ethylene flame. Ethylene is often used as a surrogate gas to represent biomass volatile gases as it has similar properties [15]. Lignell et al. [16] profiled an ethylene jet flame in great detail using a direct numerical simulation, including a temperature profile set against mixture fraction, the fraction of total mass originating from the fuel. In the flamelet simulation, mixture fraction was set to linearly vary from 1 to stoichiometric (0.122) along the fuel-rich zone, then from stoichiometric to 0 along the reaction zone and into the atmosphere. The temperature profile from ethylene jet was then multiplied by the ratio of wood adiabatic flame temperature (2253 K) over that for ethylene (2616 K) to obtain the temperature profile for the flamelet. An example, the profile for a 1 m flamelet (2 m simulation) can be seen in Figure 2.

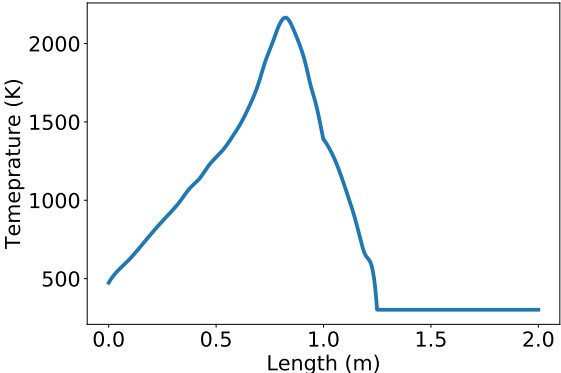

**Figure 2.** Temperature profile of a 1 m flamelet.

A chemistry profile is obtained from the mixture fraction profile. The elemental composition of the fuel was assumed to be $C_{33}H_{48}O_{19}$ [17], and these elements are mixed with air according to the mixture fraction. A gas-phase chemistry mechanism developed by Appel et al. [18] is applied to the mixture and equilibrated according to temperature and pressure (1 atm) across the entire flamelet. The result is a chemical concentration of 95 gaseous species across the entire domain, an example of which is shown for a 1 meter flamelet in Figure 3. This figure shows the partial pressure of species most relevant to the soot formation process normalized to the maximum of each individual species.

As solid fuel combusts, it first undergoes primary pyrolysis, a thermochemical conversion of solid fuel to combustible volatile species. To model this primary pyrolysis and obtain an initial yield of soot precursors, we used the biomass adaptation of the coal percolation model for devolatilization (CPDbio) [19]. CPDbio is a network devolatilization model which predicts behavior of primary pyrolysis. Inputs for CPDbio are a fuel heating profile, pressure, and mass fraction of biomass components in the fuel (cellulose, hemicellulose, and lignin). A fuel-heating profile was obtained from the computed gas temperature profile as exampled in Figure 2 for a 1 m flamelet. Pressure was assumed to be atmospheric (1 atm) throughout the simulation. Biomass components fractions were taken from a standard softwood [17] and are seen in Table 1. Outputs of CPDbio include a sectional molecular size distribution of tars. Tars represent hundreds of possible heavy species that if cooled would condense. It is these tars that serve as the primary soot precursor in the detailed soot model applied to these simulations, and CPDbio predicts tar initialization from primary pyrolysis.

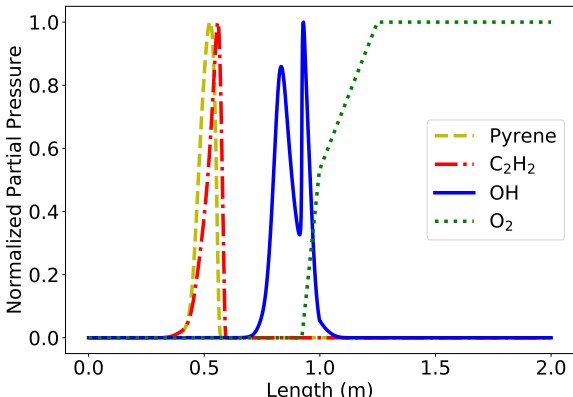

**Figure 3.** Normalized chemistry profile of a 1 m flamelet.

**Table 1.** Mass fraction of biomass components used in flamelet simulations.

| Cellulose | Hardwood | | Softwood | |
|---|---|---|---|---|
| | Hemicellulose | Lignin | Hemicellulose | Lignin |
| 0.46 | 0.06 | 0.00 | 0.21 | 0.27 |

Across this numerical flamelet, with an established precursor initialization, chemistry profile, and temperature profile, we applied a detailed soot model [14]. This soot model computes the evolution of the precursor distribution through a sectional representation (10 sections) with treatment for the physical processes of precursor growth, thermal cracking, oxidation, and soot nucleation. The evolution of the soot particle distribution is accomplished through the method of moments [20] (6 moments) with treatment for the physical processes of soot nucleation, surface growth, agglomeration, and oxidation. Precursors and soot particles were pushed through the flamelet with a constant gas velocity and pressure (1 atm). Evolution of the soot profile through the flamelet and through an agglomeration regime beyond using the detailed soot model constituted the flamelet simulations, and results generated from these simulations generated data to which an easy-to-evaluate surrogate model was calibrated.

### 2.2. Surrogate Model Development

The simulation from Section 2.1 was carried out 1200 times, with varying flame length, gas velocity, and oxygen depletion, to generate data for surrogate model calibration. Flame length is a measure of distance between fuel and the end of the reaction zone. Gas velocity is taken as the magnitude of the three velocity components in a cell. Oxygen depletion is measured on a linear scale from 0 to 1, where zeros represent no oxygen in a cell's atmosphere and 1 means the air of a cell contains 0.21 mole fraction $O_2$. From a simple QUIC-Fire grass-fire simulation, we found distributions for the flame lengths, gas velocities, and oxygen depletion encountered by a fire during a simulation. These distributions were used along with a numerical randomizer to sample flame lengths, gas velocities, and oxygen depletions for the flamelet simulations.

The results of these 1200 simulations are summarized in Figure 4. On its diagonal, this figure shows the final yield of emitted soot as a mass percentage of fuel consumed plotted against the three flamelet simulation input parameters. The off-diagonals show a two-parameter correlation plot of the soot yield to the input parameters. The figure shows the strongest correlation between flame length and yield with a somewhat weaker correlation between gas velocity and yield. The weakest indicator of yield seems to be oxygen depletion, except in cases where oxygen depletion is severe.

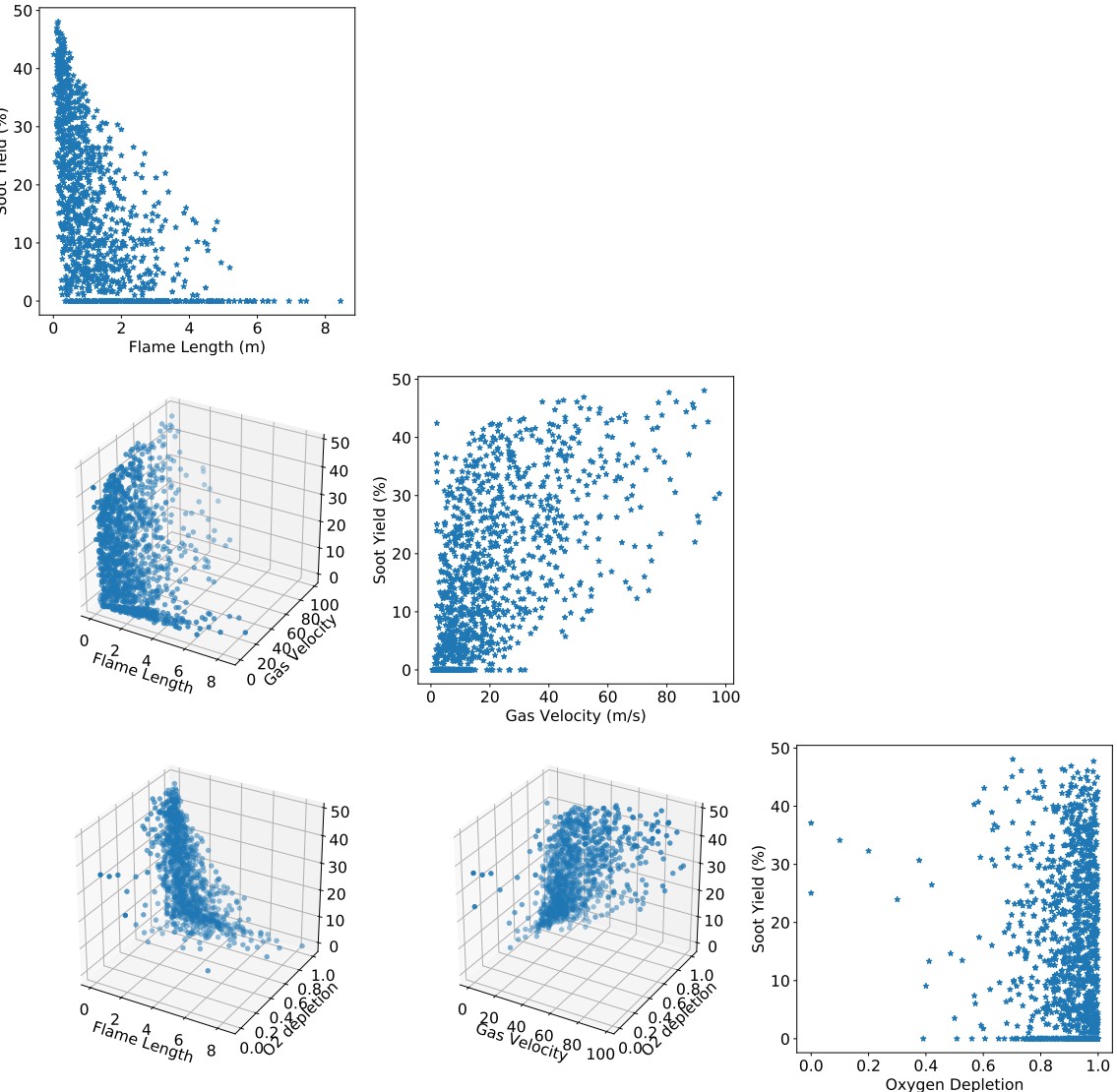

**Figure 4.** Visual summary of 1200 flamelet simulations performed where flame length, gas velocity, and oxygen depletion were varied and the ultimate yield of emitted soot, as mass percentage of the original parent fuel, was measured.

These data were used to calibrate the 2nd-degree rational surrogate model

$$Yield = SP - \frac{(aL_F + bV_G + cD_O + d)^2}{(eL_F + fV_G + hD_O + i)^2} \tag{1}$$

where $L_F$ is the flame length, in meters, $V_G$ is the gas velocity, in meters per second, and $D_O$ is the oxygen depletion, scaled from 0 to 1 as described above. $SP$ is a 'sooting potential' parameter—it is an estimate of the mass percentage of fuel converted to tars, the primary soot precursor, during primary pyrolysis. In this work, $SP$ is given a fixed value of 0.53 based off an average result of CPDbio executed over the 1200 flamelet simulations. From Equation (1), 20 tunable parameters are produced, resulting from the square of the numerator and denominator, and are calibrated using the flamelet simulation data. Once these parameters are calibrated, negligible terms are eliminated. A negligible term is defined in this work as one whose elimination does not effect the results of surrogate model by more

that 0.5% over the domain of tested variables ($0 < L_F < 20$, $0 < V_G < 100$, $0 < O_D < 1$). This calibration and elimination of negligible parameters results in a final surrogate model

$$Yield = 0.53 - \frac{2.01L_F}{2.86 + 3.48L_F + 0.162V_G - 6.77D_O + 3.95D_O^2 - 0.0634V_GD_O} \pm 0.03 \tag{2}$$

which is then implemented in QUIC-Fire. The uncertainty of this model ($\pm 0.03$) was derived from a rigorous statistical analysis described in Appendix A.

Figure 5 is a parity plot showing the agreement between the flamelet simulation data, plotted against the x-axis, and the surrogate model, plotted against the y-axis. The 45° line represents a perfect agreement between the data and model. With a $R^2$ value of 0.9897, there is impeccable agreement between model and simulation data. Initially, while performing this analysis, we thought the flame length and gas velocity parameters could be combined to a single residence time, but we found that we could not obtain the same fit using the residence time as we could with the parameters separate. Upon further investigation, we discovered the evolution of soot precursors in the flamelet simulations was weakly linked to gas velocity but was completely independent of flame length, hence the difficulty in combining the two parameters into one in the surrogate model.

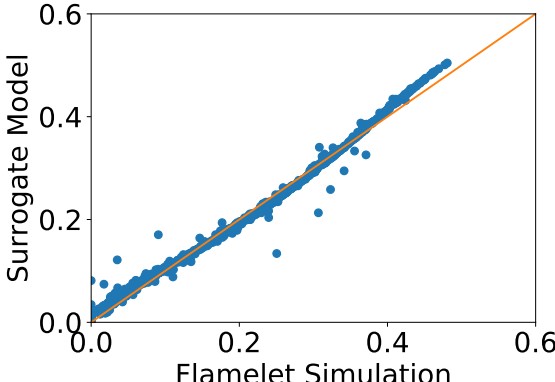

**Figure 5.** Parity plot showing the agreement between the flamelet simulation data and the surrogate model for soot mass yield ($R^2 = 0.990$).

Outputs of the flamelet simulations include 6 statistical moments of the soot particle size distribution. Using the first and second moments, we can reconstruct a lognormal distribution, a commonly observed distribution for emitted soot particles [21]. The lognormal distribution is defined by two parameters

$$f(d_p) = \frac{1}{d_p\sigma\sqrt{2\pi}} \exp\left(-\frac{(\ln(d_p) - \mu)^2}{2\sigma^2}\right) \tag{3}$$

where $d_p$ is the collision diameter of the particles and $\mu/\sigma$ are the mean/standard deviation of the distribution's exponent. Using the flamelet simulations and the above method, we were able to derive surrogate models to predict $\mu$ and $\sigma$, as well. Both surrogate molecules began as 2nd-order polynomials

$$\mu \text{ or } \sigma = (aL_F + bV_G + cD_O + d)^2 \tag{4}$$

and were calibrated with subsequent negligible parameters eliminated. The resultant surrogate model for $\mu$ is

$$\mu = -14.0 - 0.169L_F + 0.720D_O - 0.0000903V_G^2 - 0.356D_O^2 + 0.00562L_FV_G - 0.286L_FD_O \pm 0.5 \tag{5}$$

with an agreement shown in the parity plot of Figure 6. The fit of this surrogate model is not as close as that of the yield surrogate model but still effective with a $R^2$ value of 0.675.

The resultant surrogate model for $\sigma$ is

$$\sigma = 1.10 + 0.0324 L_F - 0.920 D_O + 0.513 D_O^2 + 0.0237 L_F D_O \pm 0.1 \tag{6}$$

with an agreement shown in the parity plot of Figure 7. The fit of this surrogate model is worse than that of the other two but still effective with a $R^2$ value of 0.558. In addition, Figure 7 shows there is little variation in $\sigma$ across the flamelet simulations, with over 95% of data values between 0.675 and 0.900; because of this small variation, a close fit is not as imperative to the resultant distribution as long as the surrogate model shows results within that same range, which it does.

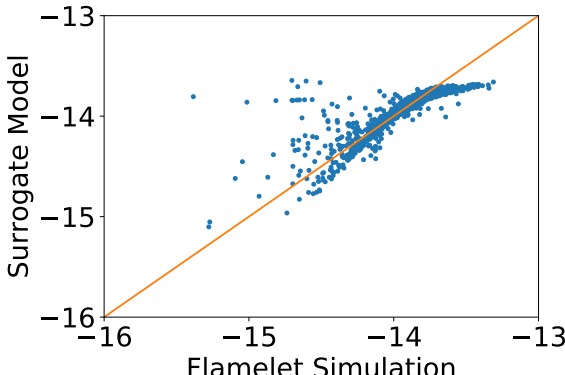

**Figure 6.** Parity plot showing the agreement between the flamelet simulation data and the surrogate model for soot size distribution $\mu$ parameter ($R^2 = 0.675$).

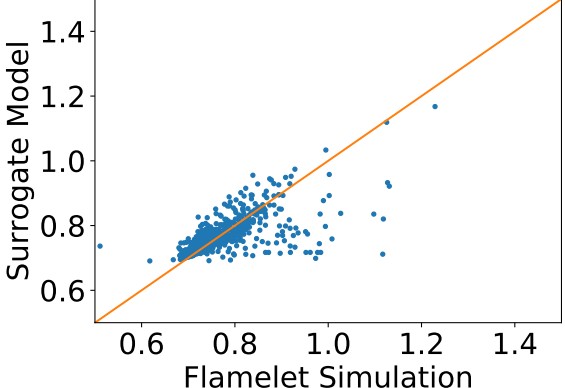

**Figure 7.** Parity plot showing the agreement between the flamelet simulation data and the surrogate model for soot size distribution $\sigma$ parameter ($R^2 = 0.558$).

## 3. Simulations

In the following sections, QUIC-Fire simulations were carried out with the proposed particle emission source term surrogate models implemented.

### 3.1. QUIC-Fire

The Quick Urban and Industrial Complex (QUIC) Dispersion Modeling System is a fast response urban dispersion model that was developed to run on desktop computers and laptops to simulate chemical, biological, and radiological agent dispersion in complex environments [22]. QUIC comprises a 3D wind field model called QUIC-URB, a transport and dispersion model called QUIC-PLUME, a pressure solver, QUIC-PRESSURE, and a graphical user interface called QUIC-GUI. To develop a fast-running fire model, the QUIC-URB diagnostic wind solver was extended to include two additional

components: a fire spread module and a buoyant plume rise formulation to be come QUIC-Fire. Note that both components are embedded in the QUIC-URB wind solver code so that feedback between the models is done online at a high temporal resolution.

Within a QUIC-Fire simulation, as fire consumes fuel, part of the released energy is used to propogate by pyrolyzing new fuel, part is lost as thermal radiation, and another part warms the air surrounding the fire. Hot air tends to rise due to buoyant forces and for a surface fire, fresh air is pulled in from the sides of the fire to compensate for the low pressure left behind by the updraft. For a single fire, fire–atmosphere interaction would generate a horizontal air motion convergent toward the fire. This is what the literature refers to as fire-induced weather [23]. The buoyant plume rise formulation predicts how the hot plumes generated by the fire move upwards and with the wind [24–26]. As plumes rise, they entrain fresh air and slow down their vertical motion. The result is a bent-over plume, with a vertical trajectory close to the fire and an almost horizontal trajectory downwind for neutral atmosphere. The plume size and vertical acceleration are computed along its trajectory and depend on the local wind conditions. Multiple plumes may interact; for instance, two perfectly vertical nearby plumes have trajectories which will bend toward one another. In particular, the plume with the smaller buoyancy will bend more toward the one with higher buoyancy [27]. Two plumes are merged when their trajectories are closer than the sum of their radii, i.e., when the plumes start to overlap. To compute the effect of the buoyant plumes motion on the winds, all the QUIC-URB cells within the plumes are assigned a vertical wind component proportional to the plume centerline updraft and the distance from the plume centerline. The modified wind field is then provided to the QUIC-URB wind solver to produce a mass-consistent fire-induced wind field that can be used to predict subsequent fire spread and emission transport and dispersion. For the wildfire simulations shown later, the typical time step is 1 s and at every time step, the wind is updated with the fire-induced air motions. This allows generating fire-spread patterns that are consistent with actual fires, e.g., the spear-shaped fire front that develops downwind of a line fire ignition in uniform surface fuel [28].

*3.2. Implementations*

To validate the application of these surrogate models, an ensemble of simulations were carried out and compared to data collected by Urbanski et al. [29]. In that work, Urbanski et al. reported fire weighted average emission factors for several species and dozens of fires. Fires were classified by type and geography, and each fire class was given a rough distribution of expected emission factors. From these data, emission factors for $PM_{2.5}$ from three classes were extracted for validation of the proposed surrogate models.

The three classes extracted from Urbanski et al.'s work were: (1) grasslands and shrublands of southeastern and midwestern United States (Grassland), (2) conifer forests of southeastern United States (SE US Conifer Forest), and (3) interior west mountain conifer forests of United States and southwestern Canada (W Mnt Conifer Forest). For each of these fire classes, a representative fuel map was obtained, and an ensemble of simulations was carried out with varied environmental conditions to try to obtain possible variances experienced by the experimentalists.

Every simulation was executed using domains shown in Figure 8. On the left is a wind domain size of $400 \times 400 \times 150$ m discretized into $2 \times 2$ m in the horizontal and with 23 stretched cells in the vertical with a surface cell depth of 1.5 m. This grid was used to simulate the winds and the buoyant plume transport. The fuel grid, on the right, was the same in the horizontal, but it was shorter in the vertical and only extended up to 23 m with a constant cell spacing of 1 m. The reason is simple: There can be no fire if there is no fuel, and in the considered simulations, the fuel was only about 21 m in height; hence, a taller grid would have been a waste of computational resources. The vertical grid spacing was also smaller to have high resolution throughout the fuel.

A logarithmic wind profile for neutral atmospheric conditions was used. The winds were updated at each time step to reflect speed up of winds where the fuel was consumed, i.e., where drag-inducing elements were removed. All simulations assumed flat topography. Total simulation time for each fire

was 800 s with a time-step of 1 s. In total, 5039 simulations were carried out. These simulations were executed using a gfortran compiler in parallel, 15 at a time on a 64-processor linux cluster that took approximately 10 days to complete all simulations.

The default fuel map for a QUIC-Fire simulation is a flat grassland field with a uniform fuel distribution and unspecified species. This default setting was used to compare against the grassland class with varying environmental conditions seen in Table 2. Variations in these and all environmental conditions were uniform across the indicated range.

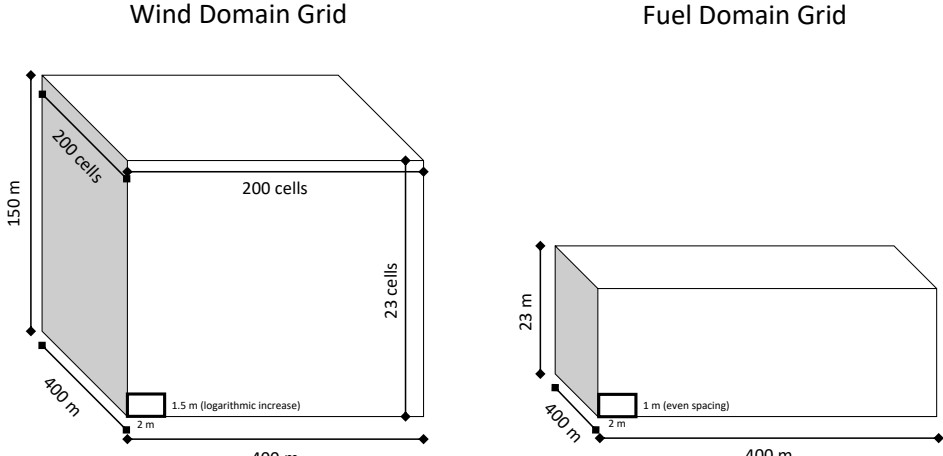

**Figure 8.** Wind and fuel grids used for the ensemble of QUIC-Fire simulations.

**Table 2.** Variations in environmental variables for each of fire class simulations.

| Fire Class | Wind Speed at 10 m (m/s) | | Initial Fireline Length (m) | |
| --- | --- | --- | --- | --- |
| | Min | Max | Min | Max |
| Grassland | 2.0 | 12.0 | 25.0 | 275.0 |
| SE US Conifer Forest | 2.0 | 12.0 | 25.0 | 300.0 |
| W Mnt Conifer Forest | 2.0 | 12.0 | 25.0 | 300.0 |
| | Ground Fuel Moisture Content (%) | | Canopy Fuel Moisture Content (%) | |
| | Min | Max | Min | Max |
| Grassland | 2 | 12 | —- | —- |
| SE US Conifer Forest | 2 | 14 | 75 | 150 |
| W Mnt Conifer Forest | 2 | 20 | 65 | 150 |
| | Ground Fuel Density (kg/m$^3$) with a Fuel Height of 0.7 m | | | |
| | min | max | | |
| Grassland | 2.0 | 12.0 | | |
| SE US Conifer Forest | 1.573 | 1.573 | | |
| W Mnt Conifer Forest | 5.5 | 5.5 | | |

To compare against the SE US Conifer Forest class, a fuel map of a regularly burned plot of land from Eglin Airforce Base, Florida was obtained, and simulations with environmental variations seen in Table 2 were carried out. Canopy fuels in this fuel map are primarily made up of long-leaf pine with lesser amounts of turkey oak and the occasional American persimmon tree. Ground fuels are primarily wire grass with a litter covering distributed according to the proximity of trees. Ground fuel density is low here because this plot is well maintained and regularly burned.

Lastly, to compare against W Mnt Conifer Forest, a closely measured stand of ponderosa pines at the Rocky Mountain Forest Service Research Center in Flagstaff, Arizona was used as a fuel map. These ponderosa pines are loosely distributed with a moderate groundcover of unspecified grass and

litter distributed according to the proximity of trees. Variations in environmental conditions are also shown in Table 2.

## 4. Results and Discussion

Figure 9 shows an example result from a grassfire simulation. Depicted in the figure is a series of images taken from the same grassfire simulation with initial conditions in the center of each value range shown in Table 2. Each image shows the computation of the particle emission source term in a single timestep. These source terms are given on a hot-cold color scale where a maximum value of 30 g of emitted particles is red and 0 g is blue. Green is the median correlating to a release of 15 g of particles in a single timestep. In this series of images, it is shown that the emission of particles increases as one moves from the head of the fire to the back. In addition, we can see that deeper fires, such as in a head fire versus a flanking fire, tend to produce more particles.

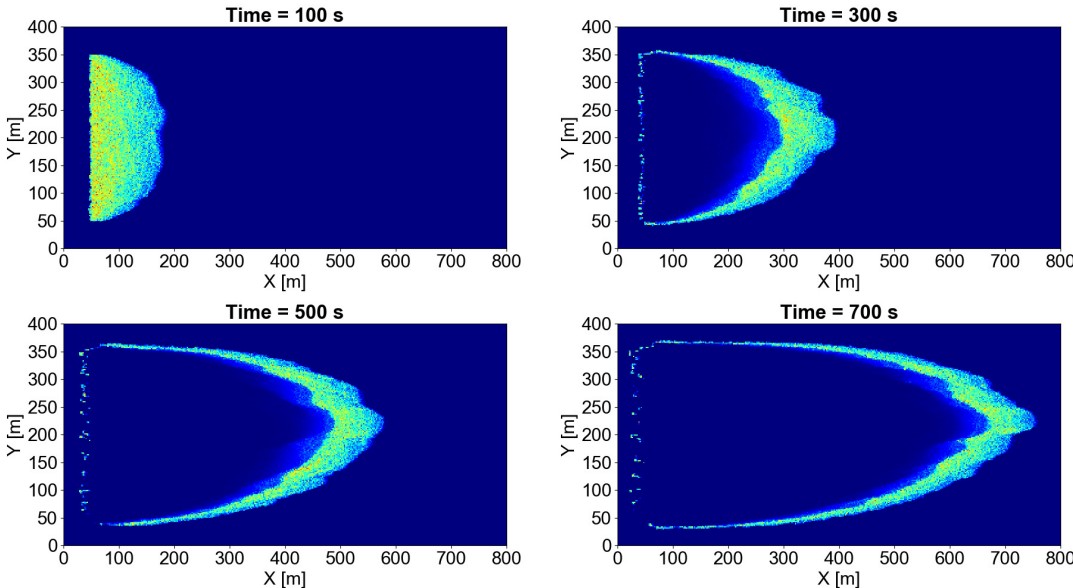

**Figure 9.** Grassfire simulation results showing the mass source term of emitted soot particles (g) in a given time-step. Each image is labeled at a timestep after ignition of and only shows the mass of emitted particles in that timestep. Color scale is maximum at red (30 g) and minimum at blue (0 g).

While the QUIC-Fire simulations also resolve some of the plume behavior and dynamics of soot particles once emitted from the fire, this study focuses on predicting the source term of these plumes at the fire level. As such, the following validation did not consider the plume dynamic portions of the above simulations but rather focused purely on the initial mass and size of produced particles at the fire level.

### 4.1. Emission Factors

Emission factors with associated quantified uncertainty for every simulation were recorded, and a mean and standard deviation of each ensemble was computed. Figure 10 shows the resulting normal distributions of emissions factor for soot particles from the simulations as a dotted line. In comparions, the normal distributions from Urbanski et al. are shown as black solid lines. In each case, there is very good agreement between the mean values of these distributions. (1) For grassland fires, we predicted a mean emission factor of 12.0 and Urbanski et al. reported 10.2, (2) for SE US conifer forest fires, we predicted 13.9 and Urbanski et al. reported 13.5, (3) and for W Mnt conifer forest fires, we predict 16.7 and Urbanski et al. reported 15.6. The close agreement between field data and simulations with regard to these mean values shows great promise for the use of this model as a predictive model in wildfire simulations.

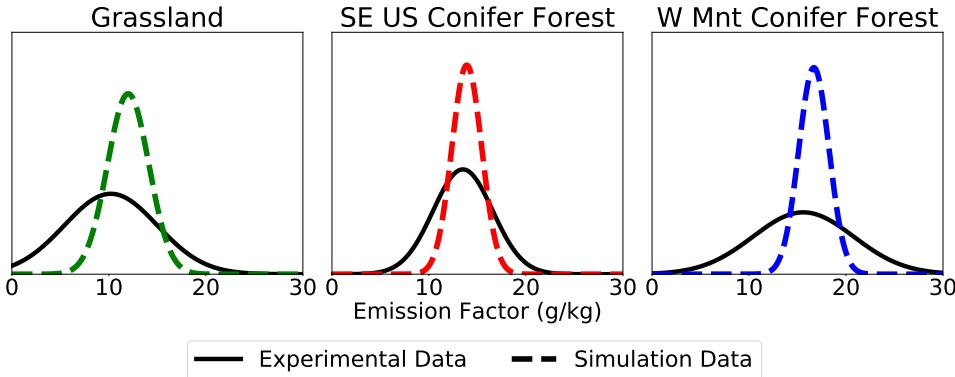

**Figure 10.** Comparison between the simulated emission factors for soot particles and emission factors for PM$_{2.5}$ from field measurements [29].

The real difference between these predicted distributions and the reported emission distributions falls in the spread of data. (1) In grassland fires, we predicted a standard deviation of emission factors of 1.58 and Urbanski et al. reported 4.80, (2) for SE US conifer forest fires, we predicted 0.53 and Urbanski et al. reported 3.07, (3) and for W Mnt conifer forest fires, we predicted 0.58 and Urbanski et al. predict 5.20. At first glance, these differences in spread are discouraging; however, one should consider the fact that Urbanski et al. are assembling data from real and different fires taken in different locations, topographies, fuel species profiles, environmental conditions, and different techniques of measurement. Naturally, this type of compilation would lead to a broad distribution of emission factors. In contrast, while we do vary the conditions shown in Table 2, the simulations are all flat with no variation in topography, and only one fuel map, with the same species and tree locations, is used for each class of fire. In addition, the method of emission factor determination is the same for each simulation as concentrations of emissions are read directly from the output data. As a result, the narrower distributions for the emission factors are not surprising, and indeed, it would have been suspicious to have similar spread considering the greater inherent variation in the ensemble of collected data compared to the simulation ensembles.

*4.2. Particle Size Distributions*

Using the presented surrogate models for the $\mu$ and $\sigma$ parameters of the lognormal distribution, an average particle size distribution (PSD) of emitted soot particles was obtained for the entire ensemble of simulations, which is shown in Figure 11. This distribution indicates that 90.4% of the total mass of particles would qualify as PM$_{2.5}$, which would shift the emission factor profiles of Figure 10 slightly to the left.

Table 3 summarizes the differences observed between simulation-averaged PSDs. To be clear, this is comparing PSD of emitted soot particles averaged across entire simulations. The minimum and maximum values of the table are not instantaneous emission profiles but still averaged over an entire simulation. The general message of Table 3 is that the predicted PSDs did not vary much from one simulation to another, which is not surprising given the tight range of values of $\mu$ and $\sigma$ shown in Figures 6 and 7, respectively.

Confidence in the predicted PSD of Figure 11 is not as high as that in the emission factor profiles of Figure 10. It is hard to compare this profile directly against literature as there is a lot of disagreement between experiments. For example, Radke et al. [30] measured particles distributions with the majority of particles found in the range between 0.1 and 1.0 µm, but with a smaller but significant mass of particles ranging between 1.0 and 10.0 µm, whereas Hosseini et al. [21] measured a particle distribution almost completely below 0.1 µm, with only a small number of particles greater that 0.1 µm and no greater than 0.3 µm.

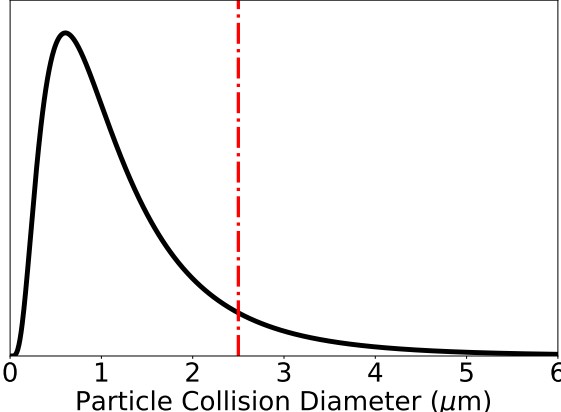

**Figure 11.** Average predicted particle size distribution of simulated fires. Lognormal distributions with $\mu$ and $\sigma$ values taken as the time-averaged values predicted across all simulations. The red vertical line represents $PM_{2.5}$.

**Table 3.** Summary of the particle size distributions resulting from the ensemble of simulations.

| Fire Class | $e^{\mu} \cdot 10^{-6}$ | | | $\sigma$ | | |
|---|---|---|---|---|---|---|
| | **Mean** | **Min** | **Max** | **Mean** | **Min** | **Max** |
| Grassland | 0.992 | 1.043 | 0.905 | 0.708 | 0.700 | 0.722 |
| SE US Conifer Forest | 1.014 | 0.900 | 1.064 | 0.704 | 0.701 | 0.715 |
| W Mnt Conifer Forest | 0.997 | 0.734 | 1.045 | 0.707 | 0.702 | 0.721 |

Compared to either case, this predicted particle size distribution is predicting particles on the large side, but not unreasonably so if we look exclusively at [30]. Perhaps the reason this model tends to overpredict particle sizes may be because the detailed soot model used to derive this surrogate model does not account for particle fragmentation, which researchers are finding plays an increasingly important role in estimating PSDs undergoing high levels of oxidation [31–33] as found in a wildfire situation.

## 5. Conclusions

Researchers have struggled with developing models to predict the formation and emission of particles in a wildfire. This work proposed a computationally efficient surrogate model for predicting a soot emission factor. The surrogate model was calibrated to thousands of more detailed simulations and is effective within the physical bounds of those simulations. When compared to experimental field data, the proposed surrogate model was surprisingly effective, although the ensemble of permformed simulations lacked the breadth of field data. The produced particle size distributions from various fires did not vary much but were reasonable. Overall, although the developed surrogate models are a rough approximation of the physics and chemistry governing particle formation and emission, those models performed well within the physical bounds to which they were calibrated. In addition, the implementation and computational execution of these surrogate models show great promise for use in low-computational-cost fire spread models.

**Author Contributions:** Conceptualization, A.J.J., T.M.H., and R.R.L.; Data curation, A.J.J.; Formal analysis, A.J.J. and T.M.H.; Funding acquisition, M.J.B. and R.R.L.; Investigation, A.J.J.; Methodology, A.J.J. and T.M.H.; Project administration, M.J.B. and R.R.L.; Resources, M.J.B. and R.R.L.; Software, T.M.H. and S.B.; Supervision, M.B. and R.R.L.; Validation, A.J.J.; Visualization, A.J.J. and S.B.; Writing—original draft, A.J.J. and S.B.; Writing—review and editing, A.J.J., S.B., and R.R.L. All authors have read and agreed to the published version of the manuscript.

**Funding:** Funding for research that contributed to basis for this work was provided by the Defense Threat Reduction Agency of the United States' Department of Defense grant number DTRA1002725370 and by the United States Department of Agriculture's Forest Service through the Rocky Mountain Forest Research Center, grant number 17IA11221633164.

**Conflicts of Interest:** The authors declare no conflict of interest. The founding sponsors had no role in the design of the study; in the collection, analyses, or interpretation of data; in the writing of the manuscript, and in the decision to publish the results.

## Appendix A. Uncertainty Quantification

*Appendix A.1*

This section defines how the uncertainty of Equations (2), (5), and (6) was obtained. The uncertainty of the models' predictions were derived from the uncertainty of the tuned parameters. These parameters' uncertainty was obtained by

$$\frac{f(\theta) - f(\hat{\theta})}{f(\hat{\theta})} = \frac{p}{n-p} F_{p,n,0.95} \tag{A1}$$

where $f(\theta)$ is Equation (2), (5), or (6), with $\theta$ representing tunable parameters and $\hat{\theta}$ being those parameters calibrated to a least-squared sum of errors between the flamelet simulations and surrogate models. $p$ is the number tunable parameters, $n$ is the number of flamelet simulations, and $F_{p,n,0.95}$ is the statistical F-distribution resolved at the 95% confidence interval.

The 95% confidence interval, equivalent to the second standard deviation, of the surrogate models' predictions is obtained from the maximum and minimum values of the predicted response within the joint parameters' confidence region. This evaluation gives us a confidence interval for every prediction as seen in Figure A1. This Figure is a replication of Figure 5 but with the model predicted-values shown in blue, the upper confidence interval for each value shown in green, and the lower confidence interval for each value shown in red. To obtain the overal uncertainty of each surrogate model, we simply took the average range of the confidence intervals of each ensemble of model prediction, i.e., the data shown in Figure A1.

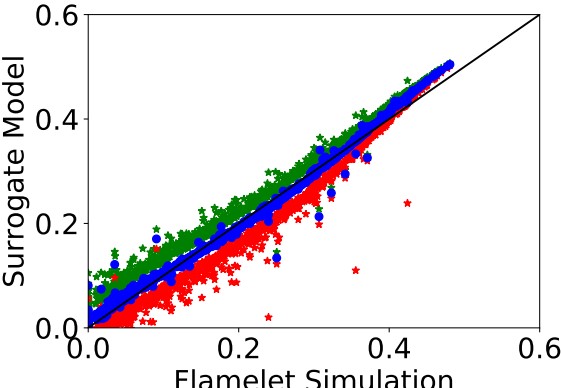

**Figure A1.** Figure 5 reproduced but with a 95% confidence interval for each surrogate prediction. Green represents the upper confidence interval and red represents the lower confidence interval.

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
