# Peer review of "Predicting Emission Source Terms in a Reduced-Order Fire Spread Model—Part 1: Particulate Emissions"

_fire, doi:10.3390/fire3010004_

Round 1
Reviewer 1 Report
This paper proposed a reduced, surrogate model to predict the particle emission source term in wildfire simulations. The surrogate model was developed using a series of 1200 flamelet simulation data points. The thus developed surrogate models, in the form of rational equations, were able to predict particle emission factor, mean, and the standard deviation of the particle sizes. To validate the surrogate models, the authors have implemented those rational equations into QUIC-fire reduced fire spread model, and carried out an ensemble of simulations for three different wildfire fires in the United States and Canada. The comparison of simulated results against the literature experimental data shows that the surrogate model is able to capture the mean emission with satisfactory accuracy, but failed to predict the its standard deviation. Overall, this paper has proposed an efficient, and reasonable accurate surrogate models for simulating soot emission behavior of wildfire. I suggest accepting this paper for publication in Fire after revisions. My comments and suggestions for improvement are listed below.
1. My largest concern is in Figure 9. The proposed model fails to capture the spread of emission factors. The authors have explained this observation to an extent. In particular, they argue that the experimental data were collected in a way that could lead to a broader distribution of emission factors. In the reviewer’s opinion, this is partially true. First, the authors did not assess how broader the distribution would be if considering the uncertainties in the experimental assembling process. At least some quantitative analysis should be present in either the main text or the supporting material. Second, as shown in Figure 7, the surrogate model performs somewhat poorly on predicting the standard deviation of soot sizes. This could be another reason for the mismatch of the spreads in Figure 9. To summarize, it would be better if the authors can add some analysis and explanation regarding the model failure on predicting the emission factor data spread.
2. The above comment leads to a further question on Figures 6 and 7. Why does the testing error so large for both the mean and standard deviation of soot particle sizes? Could it because an over parameter-elimination was carried out to derive the rational equations? Or not enough data points? The poor performance of the surrogate model could make the readers think that the model-experiment agreement of the mean emission factor shown in Figure 9 is fortuitous.
3. In Figure 2, why there is a sharp change of temperature gradient at the length of ~1.3 m?
4. The Section 3 seems a little verbal. It would be great if a schematic figure can be added in Section 3.2 to help visualize the of the computation domain (e.g. x, y, and z axis, meshes).
5. Some typos: Page 3 Line 92, it should be “pyrolysis” not “pyrolsis”; Page 8 Line 171, delete the first “will”.
Author Response
Thank you for the positive and helpful review. We, the authors, feel this work has gotten better with the response to this review. Following is the numbered comments/concerns brought up by the reviewer followed by our response in blue.
- My largest concern is in Figure 9. The proposed model fails to capture the spread of emission factors. The authors have explained this observation to an extent. In particular, they argue that the experimental data were collected in a way that could lead to a broader distribution of emission factors. In the reviewer’s opinion, this is partially true. First, the authors did not assess how broader the distribution would be if considering the uncertainties in the experimental assembling process. At least some quantitative analysis should be present in either the main text or the supporting material. Second, as shown in Figure 7, the surrogate model performs somewhat poorly on predicting the standard deviation of soot sizes. This could be another reason for the mismatch of the spreads in Figure 9. To summarize, it would be better if the authors can add some analysis and explanation regarding the model failure on predicting the emission factor data spread.
We’ve performed a rigorous statistical analysis (as much as we could do without re-running the entire ensemble of simulations which would take longer than we have to respond to this review) on the surrogate models adding uncertainties to each surrogate. Re-evaluating with these defined uncertainties broadened out Figure 9 as you’ve predicted. We’ve included now an appendix detailing how the uncertainty quantification was accomplished and updated Figure 9.
- The above comment leads to a further question on Figures 6 and 7. Why does the testing error so large for both the mean and standard deviation of soot particle sizes? Could it because an over parameter-elimination was carried out to derive the rational equations? Or not enough data points? The poor performance of the surrogate model could make the readers think that the model-experiment agreement of the mean emission factor shown in Figure 9 is fortuitous.
At this point we feel choice of surrogate model (a simple polynomial) is the limitation to a better fit to flamelet data. However, we haven’t found a better surrogate model (we’ve tried rational, logistic, exponential, and erf functions thus far) at this point. We do know that parameter-elimination and number of data points are certainly not the problems at this point.
We should note, that the particle-size surrogates (Figures 6 and 7) are independent of the yield surrogate and thus have no impact on the fortuitous fit of Figure 9 (yes, this model does better than we thought it would), rather these surrogates only affect Figure 10.
- In Figure 2, why there is a sharp change of temperature gradient at the length of ~1.3 m?
This is an aspect of our linear transitions of mixture fraction (reaction zone goes from stoichiometric to zero and the ‘atmospheric’ zone is all zero) which are an approximation of what really happens. Here is a figure of a ‘real’ flame temperature gradient, note that the ‘real’ flame does not have the sharp point our approximation does but we do feel it is a decent approximation.
- The Section 3 seems a little verbal. It would be great if a schematic figure can be added in Section 3.2 to help visualize the of the computation domain (e.g. x, y, and z axis, meshes).
Done.
- Some typos: Page 3 Line 92, it should be “pyrolysis” not “pyrolsis”; Page 8 Line 171, delete the first “will”.
Thank you, these have been fixed.
Reviewer 2 Report
The work presented deals with the developing of models to predict the formation and emission of particles in a wildfires. this is a very important topic that is not taken in account in several models. To have the possibility of predict the formation of this particles with low computational cost it is a necessary step to a better understanding and mitigation of the wildfires. In the future it would be interessant to compare with more cases field measuremnts and wildfires.
Author Response
Thank you for this positive review, we will continue to do comparisons in our own studies until we’ve developed an absolute confidence in this model but don’t really have any plans of publishing further comparisons to field measurements and wildfires (unless something really interesting surfaces.)